# Transcriptome Profiling of Maize (*Zea mays* L.) Leaves Reveals Key Cold-Responsive Genes, Transcription Factors, and Metabolic Pathways Regulating Cold Stress Tolerance at the Seedling Stage

**DOI:** 10.3390/genes12101638

**Published:** 2021-10-18

**Authors:** Joram Kiriga Waititu, Quan Cai, Ying Sun, Yinglu Sun, Congcong Li, Chunyi Zhang, Jun Liu, Huan Wang

**Affiliations:** 1Biotechnology Research Institute, Chinese Academy of Agricultural Sciences, Beijing 100081, China; joram.kiriga@gmail.com (J.K.W.); sunying0624@126.com (Y.S.); licongcong01@caas.cn (C.L.); zhangchunyi@caas.cn (C.Z.); 2Maize Research Institute, Heilongjiang Academy of Agricultural Sciences, Harbin 150086, China; cq6539@163.com; 3National Key Facility for Crop Resources and Genetic Improvement, Institute of Crop Science, Chinese Academy of Agricultural Sciences, Beijing 100081, China; sunny_5211995@sina.com (Y.S.); liujun@caas.cn (J.L.); 4National Agricultural Science and Technology Center, Chengdu 610213, China

**Keywords:** cold, stress, differentially expressed genes, transcriptome, transcription factors

## Abstract

Cold tolerance is a complex trait that requires a critical perspective to understand its underpinning mechanism. To unravel the molecular framework underlying maize (*Zea mays* L.) cold stress tolerance, we conducted a comparative transcriptome profiling of 24 cold-tolerant and 22 cold-sensitive inbred lines affected by cold stress at the seedling stage. Using the RNA-seq method, we identified 2237 differentially expressed genes (DEGs), namely 1656 and 581 annotated and unannotated DEGs, respectively. Further analysis of the 1656 annotated DEGs mined out two critical sets of cold-responsive DEGs, namely 779 and 877 DEGs, which were significantly enhanced in the tolerant and sensitive lines, respectively. Functional analysis of the 1656 DEGs highlighted the enrichment of signaling, carotenoid, lipid metabolism, transcription factors (TFs), peroxisome, and amino acid metabolism. A total of 147 TFs belonging to 32 families, including MYB, ERF, NAC, WRKY, bHLH, MIKC MADS, and C_2_H_2_, were strongly altered by cold stress. Moreover, the tolerant lines’ 779 enhanced DEGs were predominantly associated with carotenoid, ABC transporter, glutathione, lipid metabolism, and amino acid metabolism. In comparison, the cold-sensitive lines’ 877 enhanced DEGs were significantly enriched for MAPK signaling, peroxisome, ribosome, and carbon metabolism pathways. The biggest proportion of the unannotated DEGs was implicated in the roles of long non-coding RNAs (lncRNAs). Taken together, this study provides valuable insights that offer a deeper understanding of the molecular mechanisms underlying maize response to cold stress at the seedling stage, thus opening up possibilities for a breeding program of maize tolerance to cold stress.

## 1. Introduction

Maize (*Zea mays* L.) is the world’s most commonly grown cereal crop, with an estimated global annual production of about 1186.86 million metric tons in 2020/2021 [1]. The high dependence on maize for human, animal, and industrial consumption makes it one of the most critical food crops. However, maize growth and yield are highly dependent on sufficient environmental factors [2]. Thus, the current and expected scarcity of water sources and arable land due to the increasing world population and the recurrent extreme weather caused by global warming is projected to increase the incidence of abiotic stresses, such as drought, cold, and freezing during the planting, flowering, and grain-filling stages, in many corn-growing areas [3]. These abiotic stresses typically serve as crucial impediments to maize production and geographical distribution [4] and restrict agricultural yields worldwide.

Since maize has a tropical origin, cold stress is a significant risk factor among the several abiotic stresses in the development of maize. A previous report has shown that cold stress adversely affects maize growth from germination to harvest, resulting in significant yield losses due to low and slow germination and poor grain filling [5]. Corn production losses can surpass 20% in the most prolonged cold temperatures [6]. Therefore, the development of high-yielding cultivars tolerant to cold stress may help in augmenting maize production in vulnerable regions and act as an essential maize-breeding target.

The optimum maize growth temperatures range from 21 to 27 °C, while sub-optimal temperatures of about 10–20 °C decrease biomass production, thereby leading to growth retardation [7]. Cold stress induces multiple abnormalities in physiological, molecular, and biochemical processes, which harm plant growth and yield. Cell membranes may become disorganized, proteins may be denatured, oxidative defense and osmotic stress may be altered, photosynthesis possibly restricted, and metabolism may become dysfunctional, all of which subsequently disrupt growth and development, decrease fertility, and cause premature senescence and even plant death [8,9,10,11]. All of these cold stress-associated changes occur through accurate gene expression regulation and are therefore genetically regulated. Thus, screening cold stress-related candidate genes may help identify essential regulators and pathways as potential targets for breeding resistant varieties adaptable to environments with fluctuating temperatures.

As a result of their sessile nature, plants have developed complex cold acclimation mechanisms, which entail the interaction of multiple biochemical pathways in an organ-, genetic-, and environmental-specific manner [12]. They sense cold stress through changes in membrane fluidity and the accumulation of calcium signatures, leading to downstream activation of cold signaling pathways [13]. The enhanced cytosolic Ca^2+^ levels then induce C-repeat binding factors (CBFs), which act as core regulators for expressing cold-response genes [14,15]. The stress receptors, in conjunction with the cell membrane transporters, facilitate the perception of stress signals and their transmission to target genes. Multiple protein kinases, including CaMs, CMLs, CBLs, CDPKs, and MAPKs, phosphorylate other kinases and/or various TFs, resulting in activation of the cold-responsive genes [16]. Moreover, the transcriptional factor (TF) families bHLH, CAMTA, MADS, WRKY, NAC, TRAF, C3H, and AP2/ERF are critical in cold response mechanisms, while phytohormones, such as abscisic acid (ABA), regulate specific pathways that lead to cold tolerance [17,18]. Cellular redox homeostasis is protected by synthesizing defense enzymes and other antioxidant systems, while soluble sugars serve as a stabilizer of cellular components and plasma membrane [19]. Secondary metabolites, such as lignin, anthocyanin, terpenoids, chaperones, and late embryogenesis abundant (LEA), provide cold tolerance by protecting cellular components from cold-induced cellular damage [20,21]. Transporters, such as the ATP-binding cassette (ABC) transporter, play integral roles in plant growth and development, homeostasis of phytohormones, and resistance to abiotic stress [22]. Tremendous progress has been made in elucidating the mechanisms underlying cold tolerance in plants. However, the complex molecular mechanisms of cold tolerance in maize seedlings are still elusive and require comprehensive research.

Moreover, considering the genetic diversity of maize inbred lines, it will be interesting to identify the stress-responsive genes that have a consistent function in a variety of inbred lines in spite of their genetic background. Differential transcriptome analysis using RNA-sequencing (RNA-seq) approaches has recently emerged as robust, reliable, and responsive to broader levels of gene expression [23]. This effective technology makes it easier to rapidly classify stress-responsive genes and decode metabolic pathways associated with biotic and abiotic stresses [24]. Currently, little is known about the transcriptomic responses of maize seedlings to cold stress. In this study, we used RNA-seq analysis to decipher the expression profiles of the differentially expressed genes (DEGs) responsible for the contrasting cold response of 46 maize inbred lines (24 tolerant and 22 sensitive) at the seedling stage. The common cold-responsive genes were then characterized by their patterns of expression and evaluated for their functional significance. The current study provides valuable clues for the in-depth characterization of molecular responses of maize seedlings to cold stress, which could lead to effective strategies for breeding and developing cold-tolerant maize varieties.

## 2. Materials and Methods

### 2.1. Plant Materials and Treatments

The maize inbred lines were derived from the hybrid 19NL, which was provided by Heilongjiang Academy of Agricultural Sciences. The maize hybrid 19NL is a highly suitable cultivar for the spring season in northeastern China. Field experiments were conducted at open field stations during the spring maize growing season (March–June 2019) in Heilongjiang (Harbin, China). In April 2019, approximately 6000 inbred lines of 19NL were planted in rows, with 40 cm between the rows and each row containing 20 seedlings. However, in the late spring of May 2019, due to climate change, the Heilongjiang region of Harbin was affected by a cold spell of below 10 °C for more than three days. We observed that the cold spell impacted the seedlings differently, with survival rates varying from one inbred line to another. This indicated that their response to varying degrees of cold stress was different, regardless of being from the same germplasm. When seedlings had six fully expanded leaves (40 days old), all inbred lines were sampled, and 46 inbred lines with contrasting cold tolerance were selected and classified into cold-tolerant (24 lines) and cold-sensitive (22 lines) lines based on the survival rates of the seedlings, as well as a visual observation of the phenotypic changes of the leaves. The top fully expanding leaves of the 24 cold-resistant and 22 cold-sensitive inbred lines were harvested, frozen into liquid nitrogen, and later stored at −80 °C for further use.

To validate the expression of our cold-responsive DEGs, we planted B73 and CIMBL116 maize inbred lines, which have previously been reported as being cold-sensitive and -tolerant lines, respectively [25,26]. The seeds from these two maize inbred lines were provided by the Chinese Academy of Agricultural Sciences’ crop science institute). Ten seeds from each inbred line (B73 and CIMBL116) were surface sterilized with 75% (*v*/*v*) ethanol for three minutes before being rinsed three times with distilled water. Seeds were then placed between two layers of damp paper at 25 °C and left to germinate in the dark for 3 days. Uniformly germinated seeds with 2–3 cm coleoptiles were selected and sown in pots filled with peat, vermiculite, and perlite (10:1:1 by vol.). The seedlings were then grown in a growth chamber with a controlled temperature of 25/20 °C (day/night), 450 L mol m^−2^ s^−1^ light density, and a 12/12 h (light/dark) photoperiod until the third leaves were fully developed. The seedlings from the two inbred lines were divided into two groups, with the first receiving a cold stress treatment of 4 °C for 2 h followed by 25 °C for 2 days. The other group, which served as a control, was kept in the growth chamber under the same conditions as described above. The control and cold treatment samples were harvested at the same time after 2 days and immediately frozen in liquid nitrogen before being stored at −80 °C for total RNA isolation.

### 2.2. RNA Extraction, Library Construction, and Illumina Sequencing

TRIzol reagent (Invitrogen, CA, USA) was used to isolate total RNA from 24 tolerant and 22 sensitive leaf samples as per the manufacturer’s standard. The samples were treated with RNase-free DNaseI (Takara, Kusatsu, Japan) to remove the genomic DNA. The NanoDrop 1000 (NanoDrop Technologies, Wilmington, DE, USA) and Agilent 2100 Bioanalyzer (Agilent Technologies, Santa Clara, CA, USA) were used to assess RNA concentration and integrity, respectively. The cDNA libraries were constructed and sequenced using the Illumina HiSeq™ 2500 platform to generate 150 bp paired-end reads. Moreover, the above procedure was also employed to extract RNA from B73 and CIMBL116 at control and treatment levels for the purpose of qRT-PCR.

### 2.3. Reads Processing, Mapping, and Gene Expression Quantification

We used FastQC (V0.11.3) to evaluate the quality of raw reads, while Trimmomatic (V0.32) was utilized to eliminate low-quality and adapter-containing reads [27]. The Phred quality scores, including Q20 (99% base call accuracy), Q30 (99.9% base call accuracy), and the GC content of the clean data, were calculated. Consequently, high-quality clean data were used in all the subsequent analyses. The maize reference genome (B73_v4) was downloaded from the maize database (http://www.maizegdb.org/genome/genome_assembly/Zm-B73-REFERENCE-GRAMENE-4.0, accessed on 15 November 2019). All the clean reads obtained from the 46 samples were aligned to the maize B73_v4 reference genome using HISAT2 (V2.0.5) [28] with default parameters. The aligned reads were assembled into transcripts, and the transcripts from all samples were merged using Cufflinks [29]. The assembled transcripts were compared to the reference annotation by Cuffcompare. The HTSeq tool [30] was used to count the number of fragments mapped to each gene, and the transcripts per million (TPM) for each unigene as the expression level was calculated. Principal component analysis (PCA) of gene expression levels was performed to calculate the distance between samples using the clustering method. Differential expression analysis was performed using the DESeq2 R package to identify DEGs in general for the whole transcriptomes of the tolerant and sensitive samples. To estimate expression level, the DESeq2 program was used to normalize the number of counts of each sample gene using the base means. The difference was calculated, and the statistical significance was determined using the negative binomial distribution test [29,31]. Genes with *p*-value ≤ 0.05 and an absolute value of log2 fold change ≥1 or ≤−1 between tolerant and sensitive samples were considered differentially expressed.

### 2.4. Functional Annotation of the DEGs

For functional annotation, the 2237 transcripts that qualified to be our DEGs were annotated against the maize genome (AGPv4, B73 RefGen_v4) (http://ensembl.gramene.org/Zea_mays/Info/Index, accessed on 17 November 2019). In total, 1656 (74%) DEGs were annotated, and 581 DEGs were unannotated. To elucidate the function of the 581 unannotated genes, we applied several previously published procedures to identify high confidence lncRNAs [32]. Briefly, (i) unannotated DEG lengths were confirmed to be longer than 200 nucleotides for further analysis; (ii) DEGs that encode open reading frames (ORFs) of 120 or fewer amino acids were retained as lncRNA candidates; (iii) DEGs with similarity to known proteins based on BlastX against the SWISS-PROT database were filtered out; (iv) all the 581 unannotated DEGs were further evaluated using Coding Potential Calculator (CPC) (http://cpc.gao-lab.org/, accessed on 20 March 2020) [33], which assesses the coding probability of transcripts; (v) a total of 337 high confidence drought-responsive lncRNAs were obtained by comparing the output of the two procedures.

### 2.5. Gene Ontology (GO) Enrichment and KEGG Pathway Enrichment Analyses

The GO enrichment analysis of DEGs was conducted by agriGOv2 (http://systemsbiology.cau.edu.cn/agriGOv2/, accessed on 12 December 2019) [34]. Significant enriched GO terms were determined by the *p*-value ≤ 0.05 with the Fisher’s exact test and the Bonferroni multi-test adjustment. Redundant GO terms were removed using the web tool Revigo [35]. Significantly enriched GO terms were assigned to the GO categories of biological process (BP), molecular function (MF), and cellular component (CC). The KEGG (http://www.Genome.jp/kegg/, accessed on 15 December 2019) database [36] was used to analyze the functional involvement of DEGs in various metabolic pathways. Furthermore, the statistical enrichment of DEGs in KEGG pathways was tested using the KOBAS 3.0 webserver (http://kobas.cbi.pku.edu.cn/waitkobas.php, accessed on 15 December 2019) [37], while the criteria for substantially enriched KEGG pathways was a *p*-value ≤ 0.05. A co-expression network was constructed using the R package based on a weighted gene co-expression network analysis (WGCNA) to identify significant hub genes associated with cold tolerance in maize.

### 2.6. Validation of Cold-Responsive DEGs by Quantitative Real-Time PCR (qRT-PCR)

To validate the reliability and repeatability of the RNA-Seq data, six DEGs were randomly selected for verification by qRT-PCR. The gene-specific primers (Appendix A) were designed using Primer Premier 5.0 software (Premier Biosoft International, Palo Alto, CA, USA). The qRT-PCR was conducted in triplicate using 2× ChamQ SYBR qPCR Master Mix Kit (Low ROX Premixed) (Vazyme Biotechnology Co., Nanjing, China) on an Applied Biosystems QuantStudio^®^ 6 Flex (Thermo Lifetech, Waltham, MA, USA). The reaction system utilized was as previously described by Zhang et al. [38]. The internal reference β-actin was utilized to normalize the expression data. The relative expression levels of the six DEGs were calculated according to the 2^−ΔΔCT^ (cycle threshold) method [39].

## 3. Results

### 3.1. Phenotypic Analysis of Maize Population under Cold Stress Conditions

Forty-six inbred lines cultivated in Heilongjiang Province, China, were selected and evaluated for cold tolerance based on their seedling survival rate and physiological response (visual observations of the leaves). From the results, 24 inbred line seedlings displayed little phenotypic changes, maintained fully expanded green leaves, had intact plant architecture, strong vigor, and high survival rates (average of 0.8) (Appendix A). These inbred lines were therefore classified as cold tolerant and were labeled with an extension of −1 (Figure 1 and Appendix A). The remaining 22 inbred line seedlings showed visible phenotypic damage, including shriveled, curled, and yellowish spots on the leaves, and low survival rates (average of 0.3) (Appendix A). These seedlings were classified as cold sensitive and were labeled with an extension of −2 (Figure 1 and Appendix A). Collectively, the susceptible lines were more severely damaged by cold stress than the tolerant lines, as evidenced by the shriveled, curled, and yellowish patches on their leaves, as well as low seedling survival rates of about 0.3 on average.

### 3.2. RNA-Seq Analysis and Alignment of Unique Reads to the Maize Reference Genome

The RNA for the RNA-seq analysis was extracted from the top fully expanding leaves of six-leaf-stage maize seedlings of the 24 tolerant and 22 sensitive maize inbred lines mentioned in Section 2.1 above. The cDNA libraries developed from the RNA described above were constructed and used for Illumina Genome Analyzer (HiSeq™ 2500) deep sequencing. The resulting raw data were deposited into the Genome Sequence Archive under accession number CRA003678 and are publicly accessible at https://bigd.big.ac.cn/gsa, accessed on 5 January 2021). After the filtration of low-quality sequences and adaptor sequences, the 24 tolerant and the 22 sensitive libraries produced 1.13 and 1.12 billion paired-end reads, respectively (Appendix A). A total of 2.247 billion paired-end reads with a length of 2 × 150 base pairs (bp) and an average of 48.84 million clean reads per library were obtained (Appendix A). The Q20 percentages (sequencing error rates lower than 1%) were more than 95.8%, while the Q30 base percentage, which is an indicator of the overall reproducibility and quality of the assay, was greater than 90%. Moreover, the GC content of each library was 52.6% on average (Appendix A). About 1.8 billion clean reads (82%) were mapped to the maize B73_v4 reference genome using HISAT2. Multiple mapped clean reads in each library were excluded from further analysis, and only uniquely mapped clean reads were used for subsequent analysis.

### 3.3. Identification, Annotation, and Differential Analysis of DEGs

The transcription level was calculated via HTSeq-count as transcript counts, and a total of 53,037 transcripts were obtained and normalized with DESeq2. The expression patterns of these transcripts were investigated in both 24 tolerant and 22 sensitive samples. A definite expression pattern of a single transcript was established after a group comparison analysis (24 tolerant versus 22 sensitive samples), and its base mean, log2 fold change, *p*-value, and padj values were acquired. From this information, a gene was considered differentially expressed if the *p*-value was ≤0.05 and the log2 fold change value was ≥1 or ≤−1 between the 24 tolerant and the 22 sensitive samples. A total of 2237 DEGs were detected in all 46 samples. A principal component analysis (PCA) plot was generated based on the gene expression levels of the 46 samples, and the PC1 and PC2 explained 22% of the total variance (Figure 2). The PCA revealed that the correlation of the 46 samples was based on their response to cold stress. Both the tolerant and sensitive samples clustered together, implying a differential response to cold stress. A heatmap of the 2237 DEGs revealed two distinctive clusters, with 1064 DEGs possessing a positive log2 fold change enriched in the tolerant samples, and 1173 DEGs with a negative log2 fold change were enriched in the sensitive samples (Figure 3). Collectively, cold stress upregulated 1064 DEGs in the tolerant lines, while it downregulated them in the sensitive lines. Similarly, cold stress upregulated 1173 DEGs in the sensitive lines, while downregulating them in the tolerant lines.

The annotation of the 2237 DEGs with the maize reference genome B73 RefGen_v4 model resulted in 1656 (74.0%) annotated and 581 (26%) unannotated DEGs. The differential analysis of the 1656 annotated DEGs was carried out based on the two categories of cold-tolerant and cold-sensitive lines. Resultantly, 779 and 877 DEGs were significantly enhanced in the tolerant and sensitive lines, respectively. Further analysis of the 1656, 779, and 877 DEGs was carried out to establish the pathways involved in the cold stress response among the 46 samples.

For the 581 unannotated DEGs, their sequences (≥200 bp) were uploaded to the CPC website for classification as protein-coding or non-coding RNAs. A total of 271 DEGs were classified as protein-coding, while 310 were classified as long non-coding RNAs (lncRNAs) (Figure 4). Moreover, the 581 unannotated DEGs were scanned for the open reading frame (ORF). A total of 402 DEGs with an ORF greater than 120 amino acids (aa) were discarded. The remaining 179 DEGs (ORF length ≤ 120 aa) were aligned to the SWISS-PROT database for the identification of homologous proteins. A total of 56 DEGs were discarded after being homologous to known proteins (E-value ≤ 0.001), while the remaining 123 DEGs were classified as lncRNAs (Figure 4). In total, 337 putative cold-responsive lncRNAs (Figure 4) were identified from the 581 unannotated DEGs, implying the role of lncRNAs in the cold stress response. We analyzed the top-most 20 DEGs regulated by cold stress in both tolerant and sensitive lines and half were unannotated (Table 1). However, most of the annotated DEGs remain uncharacterized, which implies more research is required to unravel the molecular mechanism of cold tolerance.

### 3.4. Gene Ontology (GO) Analysis of DEGs

To identify the DEGs’ significantly enriched GO terms, the functions of the 1656, 779, and 877 (46 inbred lines, 24 cold-tolerant lines, and 22 cold-sensitive lines, respectively) DEGs were analyzed using AgrigoV2 software. All DEGs were classified into three main GO categories: cellular components, molecular functions, and biological processes. The GO terms related to the response to cold (GO: 0009409), homeostatic process (GO: 0042592), response to temperature stimulus (GO: 0009266), regulation of biological quality (GO: 0065008), response to abiotic stimulus (GO: 0009628), multicellular organismal process (GO: 0032501), response to stress (GO: 0006950), G-protein-coupled receptor protein signaling pathway (GO: 0007186), transcription (GO: 0006350), and cell surface receptor-linked signaling pathway (GO: 0007166) were among the most common significantly enriched terms in the biological process (BP) category of all the three groups of DEGs named above (Figure 5). Within the molecular function (MF) category, catalytic activity (GO: 0003824), protein tyrosine kinase activity (GO: 0004713), G-protein-coupled receptor activity (GO: 0004930), water binding (GO: 0050824), and transcription regulator activity (GO: 0030528) were the most significantly enriched in all the three groups of DEGs (Figure 5). In the cellular component category (CC), integral to membrane (GO: 0016021) and intrinsic to membrane (GO: 0031224) were the common significantly enriched categories (Figure 5). The GO enrichment analysis of the tolerant and sensitive lines (Figure 6) showed that the GO term of metal ion transport (GO: 0030001) was significantly enriched in the tolerant lines (Figure 6A), while the GO terms of apoptosis (GO: 0006915), proteolysis (GO: 0006508), cell death (GO: 0008219), death (GO: 0016265), and programmed cell death (GO: 0012501) were significantly enriched in the sensitive lines (Figure 6B).

### 3.5. KEGG Pathway Analysis of DEGs

To further explore the biological pathways of the 1656, 779, and 877 DEGs involved in maize response to cold stress, we assessed the number of DEGs in each KEGG pathway. For the 1656 DEGs, the KEGG pathways of lipid metabolism (linoleic acid and arachidonic acid), biosynthesis of other secondary metabolites (isoquinoline alkaloid, betalain, and phenylpropanoid biosynthesis), signal transduction (MAPK signaling pathway-plant), amino acid metabolism (alanine, aspartate, glutamate, glycine, serine, threonine, and phenylalanine metabolism), membrane transport (ABC transporters), terpenoids and polyketide metabolism (carotenoid biosynthesis), and metabolism of other amino acids (glutathione metabolism) were significantly enriched (Figure 7). For the 779 DEGs obtained from the cold-tolerant lines, lipid metabolism (linoleic acid, α-linolenic, ether lipid, arachidonic acid, and glycerophospholipid metabolism), replication and repair (base excision repair), biosynthesis of other secondary metabolites (monobactam and phenylpropanoid biosynthesis), membrane transport (ABC transporters), metabolism of terpenoids and polyketides (carotenoid biosynthesis), and metabolism of other amino acids (glutathione metabolism) were the most significantly enriched pathways (Figure 8A). Within the 887 DEGs from the cold-sensitive lines, the KEGG pathways of biosynthesis of other secondary metabolites (isoquinoline alkaloid biosynthesis), signal transduction (MAPK signaling pathway-plant), transport and catabolism (peroxisome), translation (ribosome), and carbon metabolism were significantly expressed (Figure 8B).

### 3.6. Dynamic Expression of Signaling and Transcription Factors Genes in Response to Cold Stress

Our GO analysis highlighted a significant number of GO terms related to signaling, such as G-protein-linked receptor protein signaling pathway, cell surface receptor-linked signaling pathway, and protein amino acid phosphorylation (Figure 5 and Figure 6). However, KEGG pathway analysis highlighted the MAPK signaling pathway-plant as one of the most significant pathways (Figure 8B). Stress sensing and signal transduction form crucial adaptive mechanisms in the tolerance of abiotic stresses. Cold stress causes a change in membrane fluidity and cytoskeleton rearrangement, thereby producing signals perceived in the cell membrane by either G-protein-coupled receptors (GPCRs) or osmotic sensors. In this study, 27 DEGs (11 enhanced and 16 suppressed) encoding GPCR were regulated by cold stress (Appendix A). Activation of the sensors leads to the generation of reactive oxygen species (ROS), plant hormonal signaling, and cell wall integrity sensing (CWI) [40]. A substantial number of protein kinases, including 21 (11 enhanced and 10 suppressed) leucine-rich repeat protein kinase family proteins, 12 (6 enhanced and 6 suppressed) protein kinase superfamily proteins, 9 (2 enhanced and 7 suppressed) mitogen-activated protein kinases, 7 (2 enhanced and 5 suppressed) S-locus lectin protein kinase family proteins, and 5 wall-associated kinases (WAKs), were regulated by cold stress (Appendix A). The high expression of protein kinase suggests that cold stress is primarily regulated at the protein level.

Transcription factors play a vital role in regulating gene expression in response to abiotic stress conditions, such as cold stress. These TFs are DNA-binding proteins that interact with cis-acting elements of genes to activate or inhibit gene transcription, hence regulating plant growth and development and response to the external environment. In the present study, we analyzed for putative TFs in the 1656 common cold-responsive genes based on the 3308 maize TFs and 56 families available in the PlantTFDB 4.0 [41]. Resultantly, 147 TFs, which fell into 32 families, were enriched from our 1656 DEGs (Table 2). ERF (18.37%), WRKY (14.29%), MYB (12.93%), NAC (8.84%), bHLH (8.84%), C_2_H_2_ (5.44%) and GRAS (2.72%) were the most abundant TF families (Table 2; Figure 9). However, the expression levels of the TF families MYB, ERF, NAC, WRKY, bHLH, and C_2_H_2_ were higher in the sensitive lines than those in the tolerant lines (Figure 9). On the contrary, the TF families MIKC_MADS, CO-like, DBB, Dof, E2F/DP, GATA, GRF, LFY, SBP, Nin-like, TALE, and ZF-HD were only induced in the tolerant lines.

Additionally, GO terms, such as regulation of transcription, transcription, and transcription factor activity, which are related to the regulation of gene expression at the transcription level, were well represented in our GO analysis (Figure 5 and Figure 6). The TF families bHLH, ERF, bZIP, and WRKY were highly regulated by cold stress in the tolerant and sensitive lines (Appendix A). As observed above, DEGs encoding the TF families LFY, MIKC_MADS, GRAS, GATA, GRF, E2F/DP, TALE, and Dof were significantly induced by cold stress only in the tolerant lines (Appendix A). In-depth research is required to uncover the roles of these TFs in the response of maize to cold stress at the seedling stage. Contrary, DEGs encoding the TF families AP2, B3, HSF, and RAV were all repressed by cold stress in the tolerant lines (Appendix A). The tolerant lines were, to a lesser extent, affected by cold stress, which might be why fewer TFs were expressed. Otherwise, our results highlight the critical roles of TFs in the regulation of cold stress response in maize seedlings.

### 3.7. DEGs Related to “Response to Cold”

The GO terms of “response to cold” and “response to temperature stimulus” were well represented in our GO analysis (Figure 5 and Figure 6). These GO terms contain crucial cold-related genes, which play significant roles in the cold regulation mechanism and may contribute to the cold tolerance of maize seedlings. The expression patterns of DEGs in these GO terms (upregulated or downregulated) may shed light on the difference between the cold response of tolerant and sensitive inbred lines. Signal transduction-related proteins, such as transmembrane proteins, serine/threonine protein kinases, leucine-rich repeat receptor-like proteins, and receptor-like kinases, were regulated by cold stress (Appendix A). This highlights the critical roles of the signal transduction network in the activation of various cold-responsive genes. The involvement of TFs in regulating cold stress was highlighted by the significant expression of C_2_H_2_, ERF, HD-ZIP, MYB, ERF, MYB, and NAC (Appendix A). Antioxidant-related enzymes, such as glutathione peroxidase, thioredoxin, and peroxidase, were regulated by cold stress, suggesting the involvement of detoxification proteins in maize response to cold stress. Carotenoid related genes, such as β-carotene isomerase and β-carotene 3-hydroxylase, further highlighted the role of ABA as a key regulator of cold stress (Appendix A). Otherwise, transporter and cell surface proteoglycan-related genes were also observed in this study (Appendix A).

### 3.8. Expression Analysis of Genes Involved in Metabolism, Transport and Functional Impacts of Co-Expressed Gene Hubs

Our KEGG analysis highlighted significant regulation of multiple metabolism pathways, including lipid, carotenoid, ABC transport, and amino acid pathways, during cold stress conditions (Figure 7 and Figure 8). Cold stress activated the phenylpropanoid biosynthetic pathway, an essential way to accumulate various phenolic compounds during cold stress conditions. In total, 14 DEGs encoding this pathway were regulated by cold stress, and 11 of which showed high expression levels in the tolerant lines (Appendix A). Moreover, there were six DEGs encoding the metabolism of alanine, aspartate, and glutamate, including β-alanine aminotransferase and glutamate decarboxylase, which are vital genes in osmotic adjustment (Appendix A).

Lipid metabolism is a dynamic and complicated process involving lipid biosynthesis, transport, accumulation, turnover, and excretion, regulating the growth and tolerance of plants to different environmental stresses. In this study, 26 lipid metabolism-encoding genes were regulated by cold stress, with all of them except one being enhanced in the tolerant lines (Appendix A). Among them were allene oxide synthase (AOS), an essential gene in the biosynthesis of jasmonic acid (JA), secretory phospholipase A2 (PLA2), α/β-Hydrolases, and phospholipase C (PLC), which are critical components of the signaling cascade, and 3-ketoacyl-CoA synthase (KCS) genes, which are related to the biosynthesis of cuticular wax (Appendix A). Moreover, six and eight DEGs encoding ABC transporters and carotenoid biosynthesis, respectively, were regulated by cold stress. ABC transporters are associated with phytohormone homeostasis, while carotenoid genes are vital for the biosynthesis of ABA, an essential hormone in cold tolerance. Otherwise, cold stress also regulated glutathione- and peroxisome-related genes, which mediate the harmful effects of ROS. Detailed information about the metabolism genes can be found in Appendix A.

WGCNA has been used to dissect the abiotic stress response in plants, thereby highlighting the power of the co-expression networks to provide deep insights into these complex processes. In this study, WGCNA identified multiple significant functional gene hubs related to the cold stress response. A total of 116 critical DEGs that were enriched by GO and KEGG into signaling, TFs, response to cold, and metabolism were defined as hub genes, highlighting the importance of their regulatory impacts on the cold stress response (Appendix A, Appendix A). Thus, WGCNA elucidated the higher order relationships between genes based on their co-expression relationships and permitted a robust view of transcriptome organization in the response of cold stress. Collectively, the combination of transcriptome analysis with WGCNA represents an opportunity to achieve a higher resolution analysis that can better predict the most important functional genes that might provide a more robust bio-signature for cold tolerance in maize, thus providing more suitable biomarker candidates for future studies.

### 3.9. Validation of DEGs by Quantitative Real-Time PCR (qRT-PCR)

To confirm the reliability and validity of the RNA-seq results in maize seedlings, six genes were randomly selected to perform qRT-PCR. Of these, XLOC_018851, *Zm00001d002748*, and *Zm00001d027330* all had a positive log2 fold change between tolerant and sensitive lines, suggesting that their expression was enhanced by cold stress treatment in tolerant lines and declined in sensitive lines. In contrast, XLOC_058556, *Zm00001d024324*, and *Zm00001d024522* portrayed a negative log2 fold change between tolerant and sensitive lines, indicating that cold stress treatment increased their expression in sensitive lines but decreased their expression in tolerant lines. As a result, the fold change ratio used in this paper emphasizes the expression pattern in cold-tolerant lines, while the inverse of that ratio reflects the expression pattern in sensitive lines. We planted B73 and CIMBL116 maize inbred lines, which have previously been reported as cold-sensitive and -tolerant lines, respectively, to validate the expression pattern of these DEGs (Section 2.1 above). These two inbred lines were planted in a cold environment as well as without cold treatment (control), allowing us to compare the expression patterns of our six DEGs before (control) and after cold treatment. Resultantly, cold stress increased the expression of XLOC 018851, *Zm00001d027330*, and *Zm00001d002748* in the tolerant line (CIMBL116), while it decreased the expression of XLOC 058556, *Zm00001d024324*, and *Zm00001d024522* (Appendix A). A reverse expression pattern was observed in the sensitive line (B73) (Appendix A). Thus, the expression trend of our DEGs shown by our RNA-Seq results was in agreement with that shown by the qRT-PCR analyses.

Furthermore, the fold change ratio of the six DEGs between the control and the cold-treated samples was calculated and compared to the fold change obtained from RNA-Seq. As a result, the cold-sensitive line’s RNA-Seq and qRT-PCR results showed an inverse expression pattern (Figure 10A) because the fold change ratio highlighted in this study emphasizes the expression pattern in cold-tolerant lines. However, if you take the inverse of that fold change ratio, which will highlight the expression pattern in cold-sensitive lines, then the RNA-Seq and qRT-PCR of the sensitive line will have a similar expression pattern to that of the tolerant line (Figure 10B). For example, cold stress enhanced the fold change of XLOC_018851, *Zm00001d002748*, and *Zm00001d027330* and declined that of XLOC_058556, *Zm00001d024324*, and *Zm00001d024522* in the tolerant lines (Figure 10B). However, cold stress inversely regulated the fold change of the above-named DEGs in the sensitive lines (Figure 10A). These results validate the authenticity of the DEGs obtained in this study, as the relative fold change in qRT-PCR matched the RNA-Seq results, implying that transcript identification and abundance estimation were remarkably precise. Furthermore, the DEGs identified in this study are universal in maize seedlings of various genetic backgrounds in response to cold stress, according to the qRT-PCR results.

## 4. Discussion

Maize has surpassed rice and wheat as the world’s most significant cereal crop. However, cold stress affects maize at any stage of development, including the germination, vegetative, and reproductive stages. Screening for cold-tolerant maize cultivars, such as rice, is difficult due to the lack of linkage between cold resistance and developmental periods [42]. Furthermore, cold tolerance is a quantitative trait influenced by the interactions of numerous genes as well as the environment. Nevertheless, RNA-seq research, which evaluates the main genes and regulatory pathways at the transcriptome level, has been widely used to investigate the molecular basis of maize response to abiotic stress [43,44]. To gain a deeper understanding of maize’s cold stress tolerance mechanisms and to develop cold-tolerant maize cultivars, we conducted a comparative transcriptome analysis of 24 cold-tolerant and 22 cold-sensitive maize inbred lines to uncover the common cold-responsive DEGs and pathways. To the best of our knowledge, this is among the first studies to profile a broad set of maize inbred lines at the seedling stage with varying levels of resistance to cold stress in the field. Previous research has usually profiled two samples with differing tolerances to specific environmental stress [45]. As a result, our study provides valuable insight into the in-depth characterization of the molecular responses of maize seedlings to cold stress, because, to date, there remains a scarcity of data on the expression patterns of critical genes and pathways across a gradient of various genotypes with varied responses to specific stress conditions.

Plants perceive abiotic stress via cell wall receptors, which activate internal signaling components through several mechanisms. The cyclic nucleotide–gated channel (CNGC) and glutamate receptors (GLRs) are the primary cell membrane cold receptors reported in plants [46]. These receptors mediate membrane Ca^2+^ fluxes and produce several endogenous signals responsible for cold tolerance [46]. Multiple G-protein-coupled receptors were shown to be strongly regulated by cold stress in our study (Appendix A). The regulation of these receptors may have caused Ca^2+^ fluxes across membranes and generated multiple signals important for maize’s cold stress response. A G-protein subunit γ gene (*Zm00001d032072*) and a trihelix GT-2 (*Zm00001d027335*) were both upregulated in the tolerant lines (Appendix A), indicating their role in cold tolerance. In a previous study, transgenic cucumber plants overexpressing *CsGG3.2* had increased CBF gene expression and were more tolerant to chilling stress [47]. COLD1 provided chilling tolerance in rice by encoding a signal regulator for guanine nucleotide-binding proteins (G-proteins) on the plasma membrane [48]. A GT2 family (*AtGT2L*) gene in *Arabidopsis* has been reported to interact with calcium/calmodulin, allowing plants to withstand cold and salt stress [49]. Two soybean GT2 genes provided abiotic stress resistance to transgenic *Arabidopsis* plants [50]. Thus, in this study, increased GPCR protein expression may have increased cytosolic calcium levels in the tolerant lines, activating a quick and diverse signaling mechanism responsible for cold acclimation.

Cold stress perceived by membrane sensors triggers an influx of Ca^2+^ into the cytoplasm, which then generates Ca^2+^ signatures that induce the activation of downstream genes, such as CBF/COR genes, in the cold signaling pathway [15]. Proteins with an EF-hand domain, such as CaMs, CMLs, CBLs, and CDPKs, act as Ca^2+^ sensors in response to cold stress [51,52,53]. Following the binding of Ca^2+^, these proteins interact with other proteins, thereby regulating downstream activities of multiple genes, which provides cold tolerance. In this study, CaMs, CMLs, CBLs, CDPKs, 21 leucine-rich repeat receptor-like kinase proteins (LRR-RLKs), 8 lectin receptor-like kinases (LecRLKs), 12 protein kinases (PKs), and 5 WAKs were regulated by cold stress (Appendix A). In *Camellia japonica*, protein phosphorylation by CDPKs and CIPKs improve cold tolerance [54], whereas in plants, WAKs play an important function in abiotic stress tolerance [55]. Interestingly, TMK1 (*Zm00001d033777*, *Zm00001d007313*), CBL10 (*Zm00001d023353*, *Zm00001d010459*), RKL1 (*Zm00001d048054*), NIK3 (*Zm00001d018635*), and PSKR (*Zm00001d018635*) displayed a high expression pattern in the tolerant lines. In a previous study on *Arabidopsis*, CBL10 mediated salt tolerance [56], while RKL1 regulated cold and salicylic acid stresses [57]. Recent research has highlighted the key roles of CBL10 in plant abiotic stress tolerance through the regulation of Na^+^ and Ca^2+^ homeostasis [58], whereas under cold stress, TMK1 significantly regulated plant development [59]. MAPKs, including MAPKK and MAPKKK, are key players in cold tolerance [48], where they phosphorylate other kinases and/or various TFs. In this study, 14 DEGs encoding MAPKs were regulated by cold stress (Appendix A). Multiple MAPKs have been implicated in improving cold tolerance in rice and Chinese jujube, according to previous research [60,61]. In this study, RBOH (*Zm00001d009248*), MAP3 (*Zm00001d001978*), and MKK3 (*Zm00001d013510* and *Zm00001d028026*) had higher expression levels in the tolerant lines (Appendix A). In a previous study, two strawberry RBOHs were reported to enhance cold stress tolerance and defense responses [62]. MKK3 was substantially expressed in tolerant lines during a comparative transcriptome investigation of two cotton cultivars with differing responses to cold [63]. The transgenic tobacco over-expressing MAP3K gene demonstrated improved tolerance to a variety of environmental stresses, including cold stress [64]. PP2C adversely controls stress-induced MAPK and SnRK2 protein kinases [4,65]. A previous study on maize has shown that stress-induced proline accumulation and tolerance to hyperosmotic stress were negatively controlled by maize PP2C [66]. This might explain why five PP2C-encoding genes were suppressed by cold stress in the tolerant lines (Appendix A). Heptahelical protein 2 (HHP2) (*Zm00001d046852*), a FASCICLIN-like arabinogalactan (FLA) (*Zm00001d016059*, *Zm00001d052819*, and *Zm00001d006009*), and a membrane-associated kinase (*Zm00001d044176*) were enhanced in tolerant lines (Appendix A). A previous study on *Arabidopsis* reported that the HHP2-MYB module is involved in integrating cold and abscisic acid signaling to activate the CBF–COR pathway [67]. Cold stress activated a membrane-associated kinase in rice, and FLAs were found to improve banana resistance to low temperatures by activating a cold signal pathway [68,69]. Collectively, the increased expression of Ca^2+^ signaling protein transcripts in maize tolerant lines activated a complex signaling cascade that regulated various downstream cold tolerance responsive genes. These genes will have significant implications for future research into maize cold tolerance at the seedling stage.

Transcription factors are key regulators of cold stress as they control multiple downstream stress-responsive genes [70]. In this study, 147 TFs belonging to 32 TF families, including AP2/ERF (27), MYB (19), bHLH (13), WRKY (21), C_2_H_2_ (8), and NAC (13), were largely regulated by cold stress (Table 2, Figure 5). Similar to our findings, in previous studies, comparative transcriptome analysis of rice, Chinese jujube, and peanut under cold stress conditions identified the above TF families to be the most regulated gene members [60,61,71,72]. These TF genes in their respective families are divided into diverse subgroups based on their specific motif structures, showing that they may perform their specific biological activities under cold stress. Moreover, individual TFs from these families have previously been reported to play a crucial function in controlling plant cold tolerance. In this study, four ERFs, namely ERF38 (*Zm00001d002748*), ERF022 (*Zm00001d048991*), DREB26 (*Zm00001d018191*), and ERF (*Zm00001d029669*), had higher expression in the tolerant lines (Appendix A). They all encode for DREB elements, which enhance cold tolerance by activating CORs. ERF38, ERF022, and DREB26 effectively regulate COR genes and sugar and proline accumulation in *Arabidopsis*, resulting in abiotic stress tolerance [73,74]. Thus, these genes may have impacted maize cold tolerance by modulating COR genes and osmotic regulators. WRKYs are yet another important class of plant TFs with diverse roles in plant response to cold stress. In this study, 18 WRKY genes were regulated by cold stress (Appendix A), with WRKY70 (*Zm00001d023332*) and WRKY53 (*Zm00001d023336*) genes showing higher expression in the tolerant lines. In previous studies, the expression of WRKY70 and *TcWRKY53* was induced by cold stress in wheat and *Thlaspi caerulescens*, respectively [75,76]. Moreover, peanut cold stress tolerance was regulated by WRKY70 and WRKY53 via the plant–pathogen interaction pathway [72]. Moreover, MYB4 (*Zm00001d041853*), bHLH57 (*Zm00001d027419*), PIF4 (*Zm00001d013130*), PTF1 (*Zm00001d045046*), AGL22 (*Zm00001d018142*), GRF6 (*Zm00001d000238*), ATHB4 (*Zm00001d002754*), Scl7 (*Zm00001d033834*), BTB/POZ (*Zm00001d023313*, *Zm00001d030864*), and C_2_H_2_ (*Zm00001d024883*) were all enhanced in the tolerant lines in this study (Appendix A). *SIPIF4* and *ZmPTF1* have been attributed to cold and drought stress tolerance in tomatoes and maize, respectively, via the modulation of ABA synthesis and signaling pathways [77,78]. In *Arabidopsis thaliana*, rice MYB4 was induced by cold stress, which in return transactivated the expression of COR genes, such as RD29A, COR15a, and PAL2 [79]. The over-expression of finger millet bHLH57 caused salinity and drought stress in tobacco [80], while *SlGRF6* was significantly regulated by cold stress in *Solanum Lycopersicum* [81]. C_2_H_2_ zinc finger proteins targeted C-repeat/DRE-binding factor genes (CBFs) to provide cold resistance in plants [82], while BTB/POZ significantly accumulated in resistant cotton cultivars during chilling stress conditions [83]. Thus, all these differentially expressed TFs identified in tolerant lines in response to cold stress could represent a useful genetic resource for breeding cold-tolerant crops. Nevertheless, many cold stress-regulating TFs are yet to be identified along with known TFs whose functions are not yet known (Appendix A). Understanding the role of the above-mentioned cold-regulating TFs at the molecular level will be pivotal in improving maize performance under cold stress conditions.

Abscisic acid is an essential plant hormone that regulates cold stress via interactions between ABA-dependent and ABA-independent pathways [84]. Moreover, exogenous ABA treatment at normal temperature improves freezing tolerance [85]. In this study, carotenoid biosynthesis genes, such as ZEP (*Zm00001d025968*), NCED (*Zm00001d042076* and *Zm00001d018819*), β-carotene isomerase (*Zm00001d007549* and *Zm00001d007560*), β-carotene 3-hydroxylase (*Zm00001d048469*), and ABA 8′-hydroxylase (*Zm00001d051554* and *Zm00001d050021*) were all enhanced by cold stress in the tolerant lines except NCED (Appendix A). In *Arabidopsis*, ABA regulates cold tolerance by improving the levels of ABA 8′-hydroxylase [86]. However, Alfalfa-related ZEP is regulated in response to drought, cold, and heat [87]. Moreover, β-carotene hydroxylase regulates the biosynthesis of a carotenoid precursor of abscisic acid called zeaxanthin. A previous report highlighted that a β-carotene hydroxylase gene caused drought and oxidative stress in rice by elevating the synthesis of ABA and xanthophylls [88]. Higher ABA levels induced cold tolerance in herbaceous plants [89]. The elevated expression of ABA-related genes was correlated to cold adaptation in a comparative transcriptome investigation of tea and tobacco plants [90,91]. Therefore, elevated carotenoid biosynthesis genes triggered ABA accumulation, and the transcriptional regulation of ABA-related gene expression is one factor that contributed to cold stress tolerance in maize. Otherwise, the suppression of NCED genes in the tolerant lines reflects the complexity of cold tolerance in plants.

Cold stress triggers rapid and intermittent ROS production that can damage plant cellular components and structures, but ROS also act as signaling molecules for abiotic stress tolerance [92]. Nevertheless, plants deploy a cascade of antioxidant machinery consisting of enzymatic and non-enzymatic defense systems to diminish the deleterious effects of ROS on plant cells [93]. The antioxidant enzymes include SOD, CAT, APX, GPX, GST, and GPX, which can trap and scavenge free radicals [94]. In this study, antioxidant genes, such as SOD (*Zm00001d014632*), GST (*Zm00001d029699*, *Zm00001d043787*, and *Zm00001d018809*), GPX (*Zm00001d029089*), PRX (*Zm00001d008266*, *Zm00001d028348*, *Zm00001d031635*, *Zm00001d032406* and *Zm00001d041827*), and APX (*Zm00001d024253*), were all enhanced by cold stress in the tolerant lines (Appendix A). The activities of SOD and GST were reported to reduce cold injury in cold acclimatized wheat [95], while the over-expression of GST in transgenic rice enhanced growth and development at a low temperature [96]. In cassava, chilling and oxidative stress was correlated with increased levels of SOD and APX genes [97]. PRXs play a role in phytoalexin-mediated plant defense and ROS metabolism [98]. Thioredoxin (TRX), however, functions as a redox transmitter [99]. Thus, the enhanced expression of TRX (*Zm00001d011352* and *Zm00001d007800*) genes in the tolerant lines might be crucial in cold acclimation through redox regulation. A previous study reported that soybean TRX genes (*Sb03g004670* and *Sb06g029490*) were significantly regulated in the cold acclimation of different accessions [100]. These findings confirm that ROS-mediated signaling could activate antioxidant enzymes, which might be responsible for imparting cold stress tolerance in maize seedlings.

The phenylpropanoid pathway and its branches of secondary metabolites are activated under cold stress, leading to the accumulation of various phenolic compounds for protection and mechanical support [101]. In this study, 14 phenylpropanoid genes, including CCR (*Zm00001d019669*, *Zm00001d008435*), PAL (*Zm00001d033286*), trans-cinnamate 4-monooxygenase (*Zm00001d016471* and *Zm00001d032468*), and β-glucosidase (*Zm00001d028199*), and 5 PRXs were significantly enhanced in the tolerant lines (Appendix A). Elevated PAL expression stimulates the biosynthesis of phenolic compounds, such as suberin and lignin, which reinforce the cell wall and prevent cell collapse during cold stress. Similarly, enhanced expression of the CCR gene has previously been correlated with lignin biosynthesis under abiotic stress [102]. Trans-cinnamate 4-monooxygenase is positioned at the turning point of phenylalanine, lignin biosynthesis, and flavonoid metabolism, making it one of the key enzymes in the synthesis of lignin and flavonoids [103]. Moreover, PRXs in the presence of H_2_O_2_ catalyze the oxidative polymerization of phenols, such as lignin precursors, which improve cell wall rigidity by boosting the cross-linking of cell wall components [104]. This increased suberin or lignin biosynthesis increases the thickness of the cell wall, preventing chilling injury and cell collapse during cold stress [105,106]. The magnitude of lignification in plants is significantly associated with their potential for cold tolerance. Previous studies have shown that β-glucosidase activates several processes, including lignin precursors [107], the release of phytohormones from inactive glycosides, and the activation of several defense compounds essential for abiotic stress tolerance [108]. However, *Nicotiana benthamiana* plants over-expressing *CsBGlu12* displayed abiotic stress tolerance via the accumulation of antioxidant flavanols that played a crucial role in scavenging ROS [109]. Collectively, the upregulation of various transcripts encoding phenylpropanoid pathway genes in the present study indicates enhanced lignification-mediated cold acclimatization in maize seedlings.

A high accumulation of osmoprotectants, such as amino acids, polyamines, quaternary ammonium compounds, and sugars, mediates diverse functions in plant defense mechanisms under varying environmental conditions [110]. Herein, genes encoding β-alanine aminotransferase (*Zm00001d038453* and *Zm00001d038460*) and glutamate decarboxylase (GAD) (*Zm00001d031749*) were enhanced by cold stress in the tolerant lines (Appendix A). The enzyme β-alanine aminotransferase catalyzes the biosynthesis of pyruvate and β-alanine, with the latter product being converted to an essential osmoprotective compound (β-alanine betaine) involved in plant abiotic stress tolerance [111,112]. However, GAD catalyzes the decarboxylation of L-glutamate to form γ-aminobutyric acid (GABA), which accumulates at high concentrations under abiotic stress [113]. Glutamate decarboxylation and GABA metabolism have been reported to play a crucial role in the cold acclimation of wheat and barley [114]. Thus, GABA played a vital role in the cold acclimation of the tolerant lines. Simultaneously, the enhanced expression of β-alanine aminotransferase facilitated a β-alanine-based osmoprotectant in maize during cold stress.

During cold stress, plants adjust their lipid content to retain membrane stability and integrity. Cold tolerance in peanuts was previously found to be associated with changes in membrane modifications, such as lipid metabolism and lipid signaling [115]. In the present study, 26 lipid metabolism genes were regulated by cold stress (Appendix A). Among them, PLC (*Zm00001d040205*), PLA2s (*Zm00001d013461*, *Zm00001d029136*), SAD (*Zm00001d024273*), AOS (*Zm00001d028282*), α/β-hydrolase (*Zm00001d010840* and *Zm00001d012147*), KCS (*Zm00001d046444* and *Zm00001d032728*), nsLTP (*Zm00001d027332*), and seven GDSL-like lipases were all enhanced by cold stress in the tolerant lines (Appendix A). AOS is a critical gene in the synthesis of jasmonic acid (JA), which affects the expression of cold-responsive genes and governs plant defense responses to various abiotic stressors [116]. In *Arabidopsis*, JA was found to provide cold acclimation [117]. PLC participates in signaling pathways that lead to the activation of the cold response through the CBF pathway [118]. The cold acclimation of spinach (*Spinacia oleracea*) leaves was found to be associated with the positive roles of PLA2 [119]. Higher expression of a SAD gene is linked to the total amount of unsaturated fatty acids (UFAs), which has been correlated to cold tolerance in tobacco plants [120]. On component change and permeability, the KCS gene catalyzes the biosynthesis of cuticular wax, which acts as a protective barrier against abiotic stresses [121]. GDSL lipases regulate plants’ development and stress response. A previous study by Kong et al. [122] highlighted an essential role of a pepper GDSL lipase gene in regulating abiotic stress tolerance. During cold stress, nsLTPs reduce lipid fluidity and membrane solute permeability, thereby reducing solute diffusion rates across the membrane and preventing osmotic membrane rupture upon thawing [123]. A previous maize study revealed that *ZmLTPs* have a role in response to cold stress [124]. Overall, our findings highlight the putative association of multiple lipid metabolic components and nsLTP proteins in maize cold tolerance.

Membrane transport systems help maintain cellular homeostasis in environmental stressful situations by redistributing different molecules, such as phytohormones, carbohydrates, and amino acids [125]. These unique roles of plant membrane transport systems may be leveraged to enhance productivity under unfavorable stress conditions as their impact on total plant physiology [126]. The increased expression of numerous transporters and channel protein genes has been reported in the *Arabidopsis thaliana* response to various abiotic stresses [127] and rice under water stress [128]. In the present study, ABCB1 (*Zm00001d024600*, *Zm00001d025703*, *Zm00001d026041*, *Zm00001d045279*, and *Zm00001d049565*), MATE efflux (*Zm00001d031730* and *Zm00001d032971*), and polyol transporter (*Zm00001d048774*, *Zm00001d029645*) genes were enhanced by cold stress in the tolerant lines (Appendix A). Plant ABCB transporters transport molecules, such as plant hormones, lipids, metabolites, contaminants, and defense molecules, which play key roles in abiotic stress tolerance. Various environmental stresses were reported to enhance the expression of distinct ABCB transporters in maize [129]. Polyols play a crucial function in the symplastic and apoplastic transfer of carbon and energy in plants’ response to salt and drought [130]. Transgenic *Arabidopsis* overexpressing the cotton MATE gene enhanced antioxidant enzyme production and abscisic acid translocation in response to cold, drought, and salt stress [131]. Therefore, the upregulation of various transporters might be associated with cold stress tolerance, the transport of plant secondary metabolites, hormones, and general growth and development in maize.

Network analysis reveals the regulatory impacts of a group of genes on target genes, revealing unique regulatory linkages that add to our understanding of abiotic stress response. The network analysis in this study had 116 nodes and 1907 connections, with 724 activation (positive) and 1183 repression (negative) connections (Appendix A). Some of the highly connected positive regulators were found among the 116 nodes, including TFs (bHLH, MYB4, MYB8, GATA4, TALE, and WRKY53) and signaling (respiratory burst oxidase, GPCR, BAM2, RKL1, NIK3, SRF7, SRF8, and PK), antioxidant (peroxiredoxin 6, peroxidase, thioredoxin), and metabolism/biosynthesis regulators (Appendix A). The cold induction of TFs regulates a set of other downstream genes. The upregulation of MYB4 (*Zm00001d041853*) in the cold-tolerant line upregulated 11 DEGs related to signaling, amino acid phosphorylation, TFs, and metabolism (Appendix A). In a previous study on *Arabidopsis thaliana*, the over-expression of OsMYB4 increased cold and chilling tolerance by increasing the expression of COR genes, such as RD29A, COR15a, and PAL2 [79]. In this study, the upregulation of the WRKY53 (*Zm00001d023336*) gene in the tolerant line increased the expression of 10 additional DEGs involved in signaling, amino acid phosphorylation, and transcription factors (Appendix A). In a previous study, WRKY53 was highly increased in *Arabidopsis thaliana* under cold stress, where it interacted with hub genes, such as mitogen-activated protein kinase 3 (MPK3), WRKY33, and WRKY40, all of which are implicated in plant defense [72]. As a result, WRKY53 could have influenced maize cold tolerance via the plant–pathogen interaction route. Differential expression levels of the 116 DEGs that make up the nodes in our cold studies show that various genes respond to cold stress in different ways and have varied biological functions. These DEGs could be intriguing candidates to investigate during maize seedling cold stress responses. Further research in this regard can look into the molecular specifics of any potential role of these DEGs in the adaptation of maize seedlings to cold stress.

Non-coding RNAs (ncRNAs), such as lncRNAs, have been discovered to regulate plant response to abiotic and biotic stress by controlling the expression of functional genes [132]. A previous study on cassava reported 318 lncRNAs responsive to cold and drought stress [133], while the expression of 2088 lncRNAs in grapevine (*Vitis vinifera* L.) was induced by cold stress [134]. In this study, the expression of 337 putative lncRNAs was regulated by cold stress (Figure 4). Thus, these lncRNAs might have modulated multiple biological processes involved in cold acclimation in maize by influencing gene expression at the transcriptional, post-transcriptional, and epigenetic levels. Otherwise, there is considerable interest in lncRNAs among molecular biologists, plant breeders, and geneticists, and our research may have identified crucial candidates that can aid in the development of cold-tolerant cultivars. However, more research is needed to fully understand the link between these lncRNAs and cold stress.

We developed a molecular model for cold stress tolerance in maize seedlings, as shown in Figure 11, based on our main findings of the critical cold-responsive DEGs and their associated pathways, as well as the numerous published citations in the present study.

## 5. Conclusions

In this study, we comprehensively compared the leaf transcriptome and phenotypic response of the maize population (24 cold-tolerant and 22 cold-sensitive lines) in response to cold stress at the seedling stage. Resultantly, the tolerant lines maintained a strong vigor with higher survival rates, while the majority of the sensitive line seedlings died and had yellow spots on the leaves. Using the RNA-seq-based approach, 2237 (1656 annotated and 581 unannotated) DEGs were identified between the tolerant and sensitive samples. Moreover, cold stress significantly enhanced 779 and 887 DEGs in the tolerant and sensitive lines, respectively. Functional annotation was carried out on the three categories (1656, 779, and 887) of DEGs. In the tolerant lines, genes associated with GPCR, Ca^2+^ signaling, protein kinases, and ROS may have played a significant role in rapid sensing and signaling, whereas genes associated with hormones, such as ABA and JA, may have played a role in signaling and cross-talk between diverse stimuli. The activation of TFs and their binding to promoter sites of certain genes results in activation of stress-responsive genes. The upregulation of several antioxidants, transport, and osmoprotectants suggested protection of the cellular machinery, whereas genes associated with the phenylpropanoid biosynthesis pathway might be involved in providing mechanical support and protection against cold stress. Moreover, genes involved in lipid metabolism may play a critical role in cold stress resistance via membrane modification. Thus, the networks involved in the function of the genes and regulators of the above-named pathways are critical in the cold acclimation of maize at the seedling stage. Moreover, genes related to ribosome, proteolysis, peroxisome, and carbon metabolism were significantly enriched in the sensitive lines. The unannotated DEGs were more inclined in the functions of long non-coding RNAs. Our findings indicate the involvement of plant signaling, transcription factors, and protective mechanisms in the molecular mechanisms underlying cold acclimation in maize at the seedling stage. Otherwise, the essential genes and metabolic pathways identified in this study may serve as valuable genetic resources or selection targets for the genetic engineering of cold-tolerant maize cultivars.

## Figures and Tables

**Figure 1 genes-12-01638-f001:**
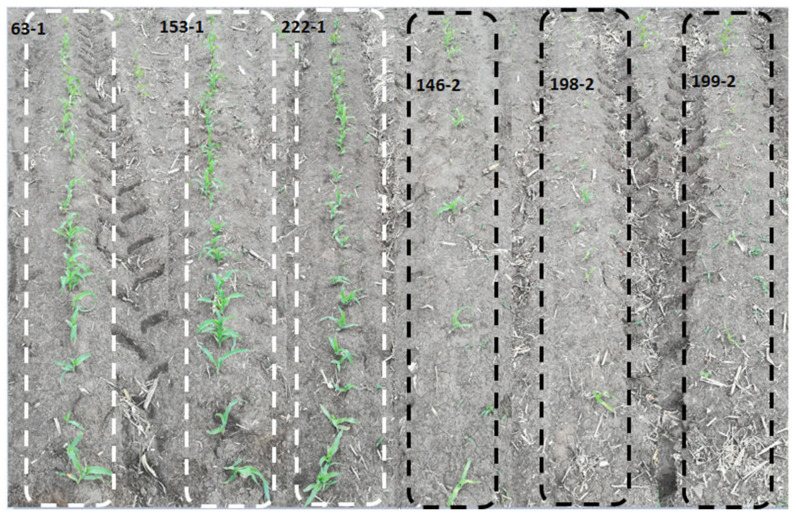
Maize seedling performance under cold stress. The performances were based on seedling survival rates and visual observations of the seedling leaves. The white and black dotted lines represent the cold-tolerant and cold-sensitive lines, respectively. The majority of the tolerant lines maintained a strong vigor with higher survival rates, while most of the sensitive line died, and their leaves had yellowish spots.

**Figure 2 genes-12-01638-f002:**
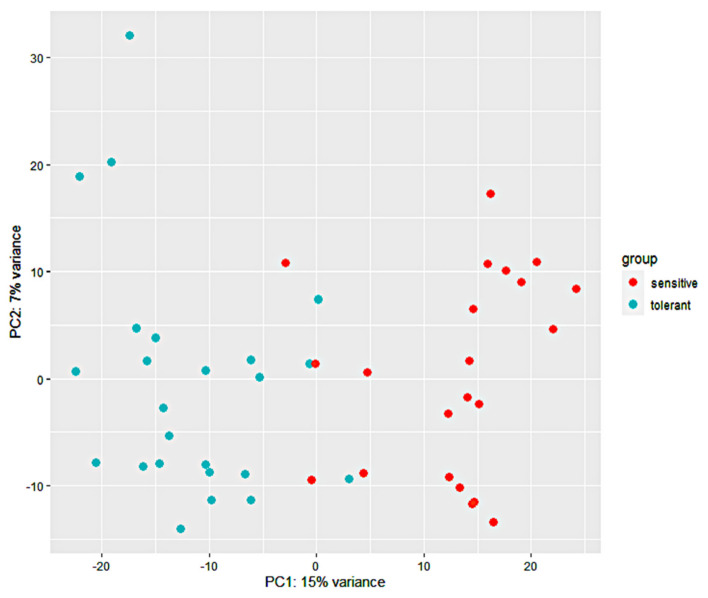
Principal component analysis (PCA) of pairwise genetic distance. The grouping of the 46 maize inbred line populations is indicated using blue (tolerant) and orange (sensitive). The proportion of variance captured is given as a percentage for both the first and second principal components (PC1 and PC2).

**Figure 3 genes-12-01638-f003:**
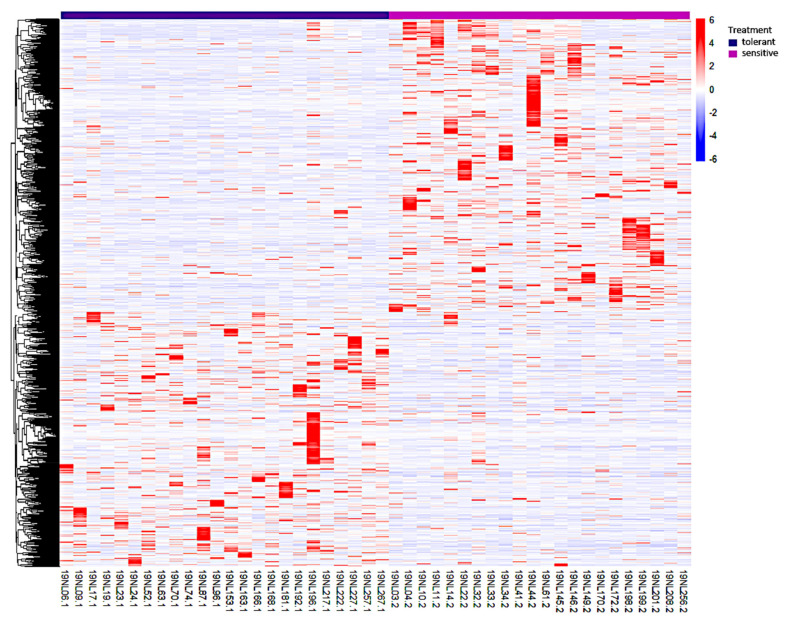
Heatmap showing the clustering analysis of 2237 common cold-responsive genes. The *x*-axis represents different maize samples. The purple color denotes the cold-tolerant lines while the pink color represents the cold-sensitive lines. The red and blue color scale represents high and low expressions, respectively.

**Figure 4 genes-12-01638-f004:**
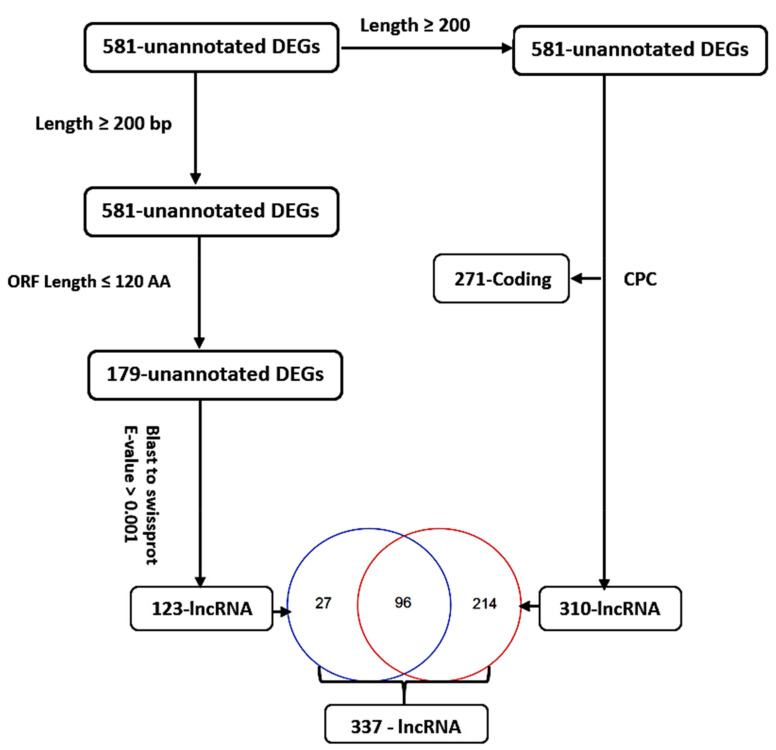
Analysis of 581 unannotated cold-responsive DEGs. CPC generated 271 coding and 310 long non-coding RNAs. The ORF method generated 123 long non-coding RNAs. In total, 337 putative long non-coding RNAs were generated by the two methods.

**Figure 5 genes-12-01638-f005:**
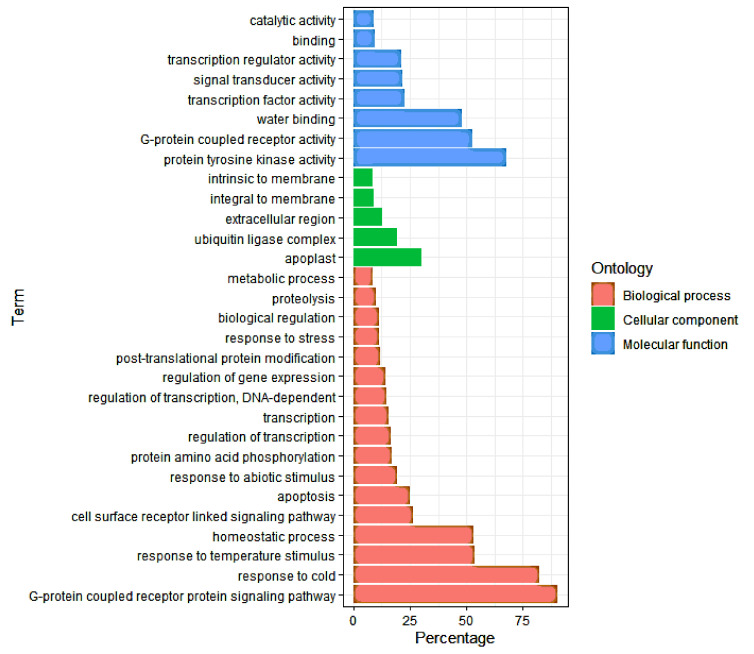
Gene Ontology (GO) enrichment analysis of the 1656 common cold-responsive genes. The GO terms shown here are the top-most biological process (BP), molecular function (MF), and cellular component (CC) categories.

**Figure 6 genes-12-01638-f006:**
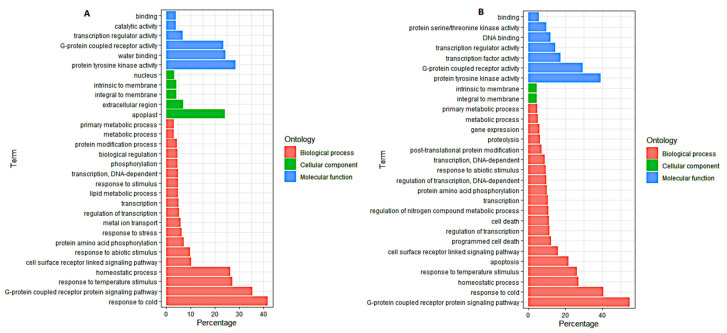
Gene Ontology (GO) enrichment analysis. (**A**) The 779 DEGs highly enriched in the tolerant lines. (**B**) The 877 DEGs highly enriched in the sensitive lines. The GO terms shown here are the top-most biological process (BP), molecular function (MF), and cellular component (CC) categories.

**Figure 7 genes-12-01638-f007:**
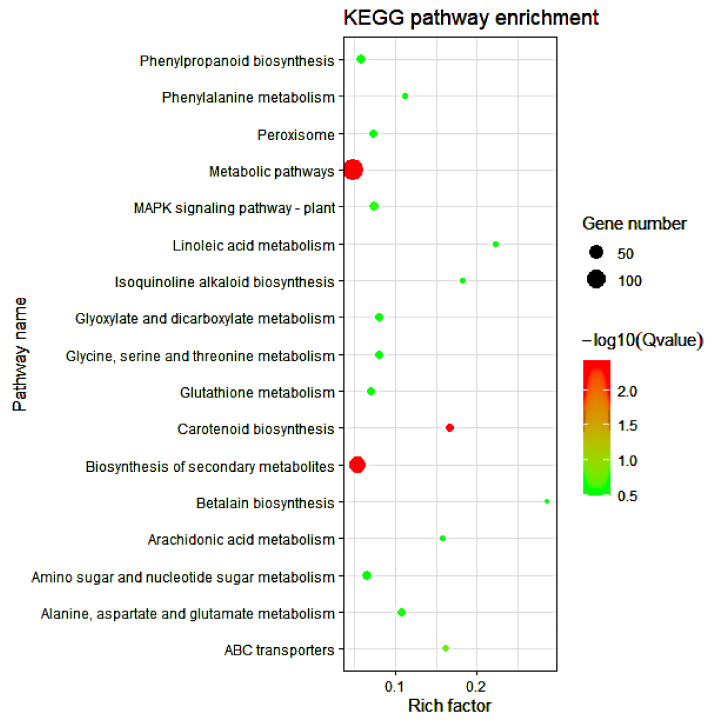
KEGG pathway enrichment analysis of the 1656 common cold-responsive genes. The experimental comparisons were based on the hypergeometric test, while the significance of the enrichment of the KEGG pathway was based on q value, q < 0.05. The color gradient represents the size of the q value; the color ranges from green to red, and the closer to green, the smaller the q value and the higher the significant degree of enrichment of the corresponding KEGG pathway. The “rich factor” represents the percentage of DEGs to total genes in a given pathway.

**Figure 8 genes-12-01638-f008:**
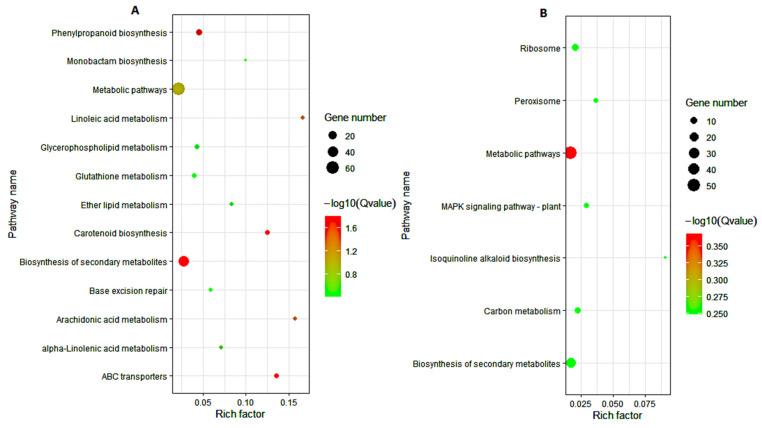
KEGG pathway enrichment analysis. (**A**) The 779 DEGs highly enriched in the tolerant lines. (**B**) The 877 DEGs highly enriched in the sensitive lines. The experimental comparisons were based on the hypergeometric test, while the significance of the enrichment of the KEGG pathway was based on q value, q < 0.05. The color gradient represents the size of the q value; the color ranges from green to red, and the closer to green, the smaller the q value and the higher the significant degree of enrichment of the corresponding KEGG pathway. The “rich factor” represents the percentage of DEGs to total genes in a given pathway.

**Figure 9 genes-12-01638-f009:**
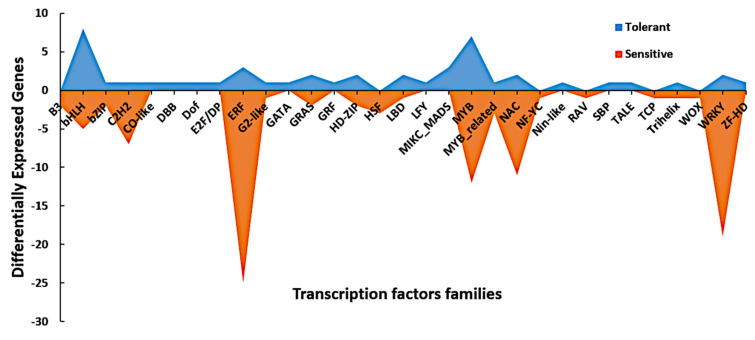
Area map of the various transcription factor families regulated by cold stress. The *x*-axis represents the TF families, while the number of genes per family is represented by the *y*-axis. The blue and orange colors indicate the TFs regulated in the tolerant and sensitive lines, respectively.

**Figure 10 genes-12-01638-f010:**
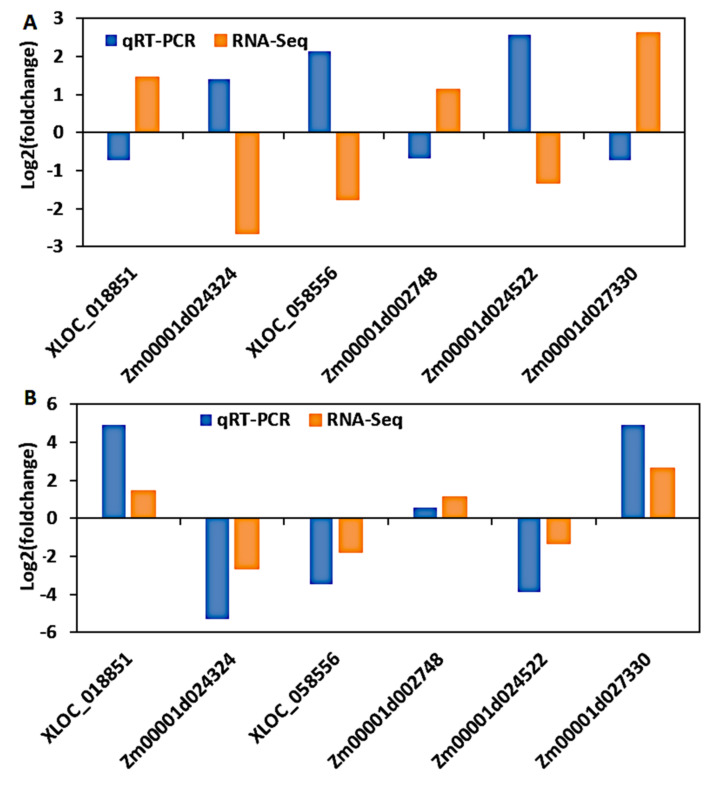
Validation of RNA-Seq results by qRT-PCR. Each log2 fold change calculated from qRT-PCR was compared with the log2 fold change of the RNA-Seq data. (**A**) The inverse expression patterns of the RNA-Seq and qRT-PCR results from the cold-sensitive line (B73) is because the fold change ratio highlighted in this study emphasizes the expression pattern in cold-tolerant lines. However, if you take the inverse of that fold change ratio, which will highlight the expression pattern in cold-sensitive lines, then the RNA-Seq and qRT-PCR expression trends would be identical. (**B**) The cold-tolerant line’s (CIMBL116) RNA-Seq and qRT-PCR results show a similar expression trend. Orange and blue bars represent RNA-Seq and qRT-PCR data, respectively.

**Figure 11 genes-12-01638-f011:**
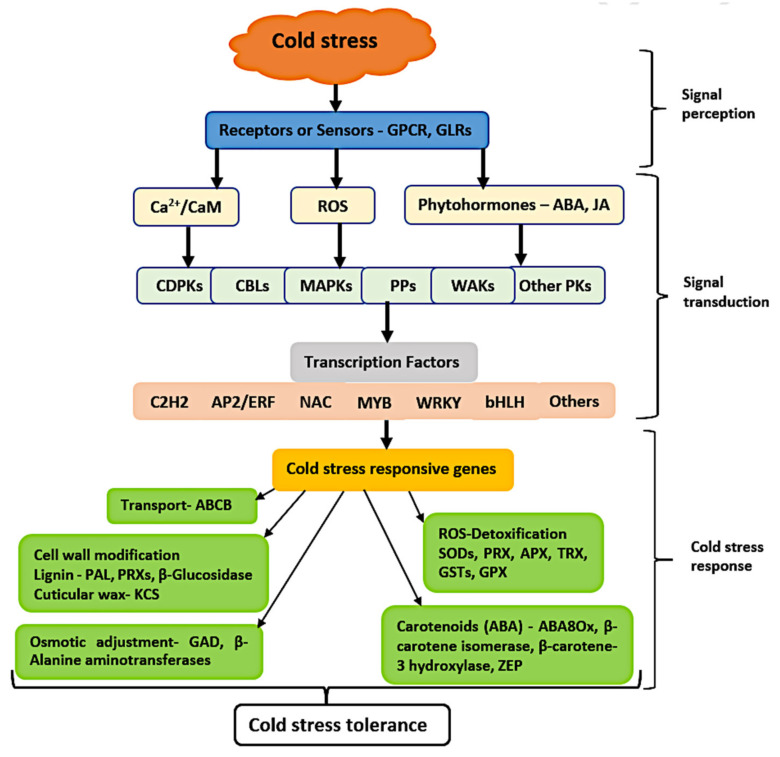
The schematic molecular model describing the signaling pathways involved in the acquisition of cold tolerance in maize seedlings. The model was constructed based on the main cold response components identified in this report, as well as plant abiotic stress pathway schemes previously described. The downward pointing arrows represent the sequence of events in cold tolerance in maize, from stress signal perception to acclimation mechanisms. Abbreviation key: GPCR, G-protein-coupled receptor; GLR, glutamate receptor; ROS, reactive oxygen species; ABA, abscisic acid; JA, jasmonic acid; CDPKs, calcium-dependent protein kinases; MAPK, mitogen-activated protein kinase; PPs, protein phosphatase; WAKs, wall-associated kinases; PKs, protein kinases; PRX, peroxidases; PAL, phenylalanine ammonia lyase; KCS, ketoacyl-CoA synthase; GAD, glutamate decarboxylase; SOD, superoxide dismutase; APX, ascorbate peroxidase; TRX, thioredoxin; GST, glutathione transferase; GPX, glutathione peroxidase; ZEP, zeaxanthin epoxidase.

**Table 1 genes-12-01638-t001:** Transcription factor gene families identified from 2237 DEGs in maize under cold stress.

TF Family	DEGs in the Tolerant Line	DEGs in the Sensitive Line	Total
B3	0	2	2
GRF	1	0	1
ERF	3	25	28
DBB	1	0	1
Dof	1	0	1
HSF	0	3	3
LBD	2	1	3
LFY	1	0	1
MYB	7	12	19
NAC	2	11	13
RAV	0	1	1
SBP	1	0	1
WOX	0	1	1
TCP	0	1	1
E2F/DP	1	0	1
GATA	1	0	1
GRAS	2	2	4
bZIP	1	2	3
C_2_H_2_	1	7	8
bHLH	8	5	13
Nin-like	1	0	1
NF-YC	0	1	1
ZF-HD	1	0	1
G2-like	1	1	2
CO-like	1	0	1
HD-ZIP	2	2	4
WRKY	2	19	21
TALE	1	0	1
Trihelix	1	1	2
MIKC_MADS	3	0	3
MYB_related	1	3	4
Total	47	100	147

**Table 2 genes-12-01638-t002:** List of top 20 most regulated DEGs by cold stress.

Locus ID	Gene ID	log2 Fold Change (T/S)	*p*_Value	Chr	Start	End	Annotation
XLOC_000983	Zm00001d027606	23.73473512	6.81 × 10^−31^	Chr1	8,915,719	8,918,057	transmembrane protein
XLOC_056725	-	−23.7343301	9.38 × 10^−29^	Chr7	170,162,017	170,164,579	-
XLOC_046285	Zm00001d017622	−2.670016539	9.74 × 10^−25^	Chr5	201,997,628	201,998,691	OSJNBa0088A01
XLOC_018249	-	24.01063195	1.05 × 10^−24^	Chr2	137,103,173	137,104,433	-
XLOC_058213	Zm00001d021006	−1.554984072	3.70 × 10^−22^	Chr7	140,181,294	140,183,018	MTD1
XLOC_034912	-	23.0239833	5.49 × 10^−21^	Chr4	176,582,912	176,584,805	-
XLOC_056365	Zm00001d021394	22.79156651	7.11 × 10^−21^	Chr7	150,891,109	150,895,395	hypothetical protein
XLOC_067464	-	20.05873411	6.49 × 10^−20^	Chr9	20,669,455	20,671,879	-
XLOC_049980	-	24.72008389	2.60 × 10^−17^	Chr6	114,599,197	114,601,220	-
XLOC_018351	-	24.4739764	5.35 × 10^−17^	Chr2	150,920,460	150,923,282	-
XLOC_055724	-	22.96632545	9.06 × 10^−17^	Chr7	95,897,276	95,897,694	-
XLOC_018159	Zm00001d004620	22.6740887	1.68 × 10^−16^	Chr2	122,442,335	122,447,585	uncharacterized protein
XLOC_004771	-	−23.94951263	2.29 × 10^−16^	Chr1	1,487,983	1,491,882	-
XLOC_007853	Zm00001d033411	−23.88106934	2.77 × 10^−16^	Chr1	262,279,726	262,300,105	hypothetical protein
XLOC_005508	Zm00001d028673	23.87942745	2.96 × 10^−16^	Chr1	42,581,898	42,586,482	small nuclear protein G
XLOC_036989	-	−23.84216727	3.11 × 10^−16^	Chr4	108,230,941	108,236,238	-
XLOC_014865	Zm00001d025968	23.6089741	6.37 × 10^−16^	Chr1	134,994,735	134,998,645	hypothetical protein
XLOC_018011	Zm00001d004338	23.58304546	6.85 × 10^−16^	Chr2	104,459,685	104,460,441	hypothetical protein
XLOC_046112	Zm00001d017287	−4.317481518	2.92 × 10^−13^	Chr5	191,750,113	191,750,625	uncharacterized protein
XLOC_027574	Zm00001d043525	−1.350259753	4.09 × 10^−13^	Chr3	202,741,852	202,743,157	oxidative stress 3

Comparison of DEGs between tolerant (T) and sensitive (S) inbred lines after cold stress. The *p*-value is less than 0.05.

## Data Availability

This manuscript includes the essential data either as figures or as Appendix A. The raw sequence reads have been deposited to the Genome Sequence Archive (GSA) under accession numbers CRA003678 (https://ngdc.cncb.ac.cn/bioproject/browse/PRJCA004123) (accessed on 5 January 2021).

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
