# Peer review of "Transcriptome Profiling of Maize (Zea mays L.) Leaves Reveals Key Cold-Responsive Genes, Transcription Factors, and Metabolic Pathways Regulating Cold Stress Tolerance at the Seedling Stage"

_genes, 2021, doi:10.3390/genes12101638_

Round 1

Reviewer 1 Report

This manuscript aims to comprehensively identify uncharacterized maize cold stress responsive genes through comparative transcriptome analysis strategy. However, several similar studies have been conducted (such as Li P, Cao W, Fang H, et al. Transcriptomic Profiling of the Maize (Zea mays L.) Leaf Response to Abiotic Stresses at the Seedling Stage. Front Plant Sci. 2017;8:290; and Jihua Mao, Yongtao Yu, Jing Yang, Gaoke Li, Chunyan Li, Xitao Qi, Tianxiang Wen, Jianguang Hu, Comparative transcriptome analysis of sweet corn seedlings under low-temperature stress, The Crop Journal, Volume 5, Issue 5, 2017), therefore, the novelty and scientific sounds of current study are limited. Below, I outlined a few of my major concerns, might help the authors to improve the MS.

  1. The study was only employed the leaf tissue for further transcriptome analysis, therefore, the title should be more precise.
  2. Which leaf did the author collect for RNA-seq analysis? Top full expanding leaf? Because many of tolerance related genes are also development dependent.
  3. The big issue of current study is lacking of control sets. We expect to investigate the expression changes compared to untreated/unstress condition. In Table 2. List of top 20 most regulated DEGs by cold stress, many genes are more than 2000 fold (Log2 > 20) changes, all of them are needed to validate by qRT-PCR.
  4. It is necessary to provide more physiological data to estimate the severity of cold treatment. Such as survival rate, ion leakage assay, ROS quantification, etc.

Reviewer 2 Report

Overall, Waititu et al. presented a good quality work on a comparative transcriptome study of maize under cold tolerance and identified a list of genes and lncRNAs associated with cold stress. I don’t have any big concerns about this paper at this stage. But there are a few suggestions I think could be further applied to increase the manuscript quality.

Comment 1: In this study, the authors identified different DEGs in different gene families and used a diagram (Figure 11) to indicate their connections. However, how are these genes associated with each other during expression and how their up or down-regulation affects cold tolerance or response are still unclear. A further co-expression network analysis and further discussion about their up/downregulation will provide insights into how the DEGs interact to respond to cold stress. This further knowledge can make this manuscript have a higher impact on the community.

Comment 2: Line 483: changed as “a broad set of maize inbreds”

Comment3: Line 696: changed as “in response to cold stress”

Round 2

Reviewer 1 Report

The authors have addressed most of my concerns. I felt that the current version should be suitable published on Plants.

Reviewer 2 Report

Happy to accept this manuscript.